# Recurrent Spliceosome Mutations in Cancer: Mechanisms and Consequences of Aberrant Splice Site Selection

**DOI:** 10.3390/cancers14020281

**Published:** 2022-01-07

**Authors:** Carlos A. Niño, Rossella Scotto di Perrotolo, Simona Polo

**Affiliations:** 1Fondazione Istituto FIRC di Oncologia Molecolare (IFOM), 20139 Milan, Italy; rossella.scotto@ifom.eu; 2Dipartimento di Oncologia ed Emato-Oncologia, Università degli Studi di Milano, 20122 Milan, Italy

**Keywords:** splicing, cancer, spliceosome, U1 snRNA, SF3B1, U2AF1

## Abstract

**Simple Summary:**

The spliceosome ribonucleoprotein complex catalyzes the removal of introns and exons ligation, a fundamental post-transcriptional process that generates mature RNAs. Cancer-associated mutations in spliceosome components give rise to aberrant splice site selection and, therefore, the production of novel isoform variants that support tumorigenesis. In this review, we summarize the current research regarding cancer hotspot mutations identified in spliceosome components acting at the very first step of splicing, namely the U1 snRNA, SF3B1, and U2AF1.

**Abstract:**

Splicing alterations have been widely documented in tumors where the proliferation and dissemination of cancer cells is supported by the expression of aberrant isoform variants. Splicing is catalyzed by the spliceosome, a ribonucleoprotein complex that orchestrates the complex process of intron removal and exon ligation. In recent years, recurrent hotspot mutations in the spliceosome components U1 snRNA, SF3B1, and U2AF1 have been identified across different tumor types. Such mutations in principle are highly detrimental for cells as all three spliceosome components are crucial for accurate splice site selection: the U1 snRNA is essential for 5′ splice site recognition, and SF3B1 and U2AF1 are important for 3′ splice site selection. Nonetheless, they appear to be selected to promote specific types of cancers. Here, we review the current molecular understanding of these mutations in cancer, focusing on how they influence splice site selection and impact on cancer development.

## 1. Introduction

Splicing is a fundamental process in gene expression regulation and is therefore essential for maintaining accurate cellular homeostasis, fitness, and fate. Splicing of precursor mRNAs (pre-mRNAs) into mature mRNAs is a two-step reaction carried out by the spliceosome, which catalyzes accurate intron removal and exon ligation [1]. The spliceosome is a ribonucleoprotein (RNP) complex composed of the small nuclear RNP subunits snRNP U1, snRNP U2, snRNP U4/U6, and snRNP U5, which associate with different splicing co-factors [1,2]. These snRNP subunits are generated through the association of the small nuclear RNAs (snRNAs) U1, U2, U4/U6, and U5 with Sm or Sm-like (LSm) proteins [2,3].

The spliceosome does not possess a preformed active site is assembled de novo directly onto each intron [1,2]. Intron removal relies on three pre-mRNA consensus sequences that are recognized by different spliceosome components: (i) the 5′ and 3′ splice sites, located at the 5′ and 3′ ends of the intron, (ii) the branch point sequence (BPS), located between 18 and 40 nucleotides upstream of the 3′ splice site, and iii) the polypyrimidine tract (PPT), located between the BPS and the 3′ splice site (Figure 1A). Spliceosome proteins are essential for the correct recognition and base-pairing between spliceosome snRNAs and the splice sites and, as such, represent an important layer of fine regulation [1,3].

During the splicing cycle, the spliceosome undergoes a series of conformational changes that are mainly represented by base pairings occurring between pre-mRNA and snRNAs [1,3], supported by DExD/H-type RNA-dependent ATPases/helicases [2] (Figure 1B).

The splicing reaction begins with the base-pair-mediated interaction between the U1 snRNA and the 5′ splice site, forming the so-called E (early)-complex (Figure 1B). At the 3′ splice site, direct RNA–protein interactions are established between splicing factor 1 (SF1) and the U2 auxiliary factor heterodimer (U2AF) with the BPS and PPT, respectively [4,5,6]. Following SF1 displacement from the BPS, the U2 snRNA pairs with the BPS, forming the pre-spliceosome or complex A (Figure 1B). U2 snRNP engagement is assisted by a direct interaction between U2 snRNP proteins and the BPS and BPS flanking regions. In particular, direct interactions are established between the U2 snRNP components p14 and SF3B1 with the adenine at the BPS and the U2AF heterodimer, respectively [7,8]. In the complex A, the interaction between U1 and U2 snRNPs brings the two splice sites close to each other, followed by the recruitment of the preassembled U4-U6-U5 tri-snRNP, forming the pre-catalytic spliceosome or complex B. This latter undergoes conformational and compositional rearrangements with the displacement of U1 and U4 snRNPs, thus determining the transition toward the complex B^act^ (activated) (Figure 1B). This complex is further remodeled into the catalytically active complex B*, which catalyzes the first transesterification reaction, generating complex C. After U2 snRNP conformational change and repositioning of the reaction intermediate, complex C catalyzes the second step of the splicing reaction. The post-spliceosome complex then dissociates, thus releasing the spliced mRNA product, the excised intron bound to U2-U5-U6, called lariat (Figure 1B), and the snRNPs that are recycled in another splicing reaction [1,2].

Aberrant splicing is commonly observed in cancer [9,10,11]. Indeed, cancer-associated alterations generate splice variants involved in all of the different hallmarks of cancer, including cancer cell survival, proliferation, metastasis, angiogenesis, and chemoresistance [12,13,14]. Aberrant splicing can arise from alterations in the components of the splicing cycle, such as mutations in splicing sites and/or spliceosome components, or from mutations or altered expression of non-spliceosome RNA binding proteins, which assist splice site selection and accurate splicing.

Three specific components of complex A have recently gained attention as they are recurrently mutated in various cancer types and affect the very first step in the splicing cycle: the U1 snRNA, which is essential for 5′ splice site recognition, the U2 snRNP component SF3B1 and the U2 auxiliar factor subunit U2AF1, both of which are involved in 3′ splice site selection [15,16,17,18,19,20]. Elegant studies that took advantage of whole-transcriptome analysis and in vivo and in vitro models have revealed the consequences of such spliceosome mutations and their impact on cancer development and progression [15,16,17,18,19,20]. Here, we review the current knowledge about the genetic characteristics of U1 snRNA, SF3B1, and U2AF1 somatic mutations and the mechanism by which they influence splicing decision and, as a consequence, tumor development.

## 2. Tumor-Associated U1 snRNA Mutations Give Rise to Aberrant 5′ Splice Site Recognition

Cancer-related alterations in the noncoding component of the spliceosome, the snRNAs, have barely been recognized and studied. One exception is represented by U1 snRNA, whose somatic mutations were recently identified in different cancer types [18]. This snRNA forms a base-pair interaction with the 5′ splice site of the pre-mRNA and is essential for the 5′ splice site recognition operated by the spliceosome [21,22] (Figure 2).

The most recurrent mutation in cancer comprises an A > G or A > C substitution at the third base of the U1 snRNA [18,19] (Figure 2). Although corresponding to the same base, each substitution has been found to be restricted to specific cancer types. The 3A > C mutation has been identified in hepatocellular carcinoma (HCC, ~6%), chronic lymphocytic leukemia (CLL, ~4%) and B-cell non-Hodgkin lymphomas (~2%) [18]. In CLL, the 3A > C mutation is restricted to the unfavorable IGHV-unmutated CLL (U-CLL) subtype and is, thus, associated with an aggressive behavior [18]. Interestingly, the 3A > C mutation was detected already at the initial stages of the disease in both CLL and HCC, suggesting that this is an early event during tumorigenesis. Instead, the 3A > G mutation has been identified in medulloblastoma (MBL, ~19%) and with a very low frequency in pancreatic adenocarcinoma (PAAD, 0.4%) [18,19]. In MBL, the U1 mutation is restricted to the sonic hedgehog (SHH) subgroup tumors and with a mutation frequency up to 50% [19]. The 3A > G mutation in SHH was found to be highly prevalent in adults (97%), less frequent in adolescents (25%) and absent in infants suffering from MBL [19]. In the case of adolescent patients presenting a mutation in SHH, the U1 snRNA mutation was associated with high risk of relapse.

### 2.1. Molecular Basis of Altered Splicing Mediated by the U1 snRNA 3A > C/G Mutant

Tumors harboring the U1 snRNA A > C/G mutation present altered splicing patterns as the mutation generates many novels 5′splice sites that are usually absent in non-mutated tissues [18,19]. Splicing alterations observed in cancer patients mainly correspond to: (i) 5′ splice site usage variations at exons and (ii) 5′ cryptic splice sites generating new exons [18,19].

The 3A > C/G mutation is located in the highly conserved 5′ splice site recognition sequence that matches the pre-mRNA [21,22] (Figure 2). The adenine in the third position directly base-pairs with the +6 base at the 5′ splice site, which is highly represented by an uracil in the 5′ splice site consensus motif (Figure 2). The 3A > C and 3A > G mutations are, therefore, predicted to impact the A:U base pairing specificity toward C:G and G:C base-pairing, respectively. Indeed, 5′ splice site which usage was increased in U1 snRNA 3A > C mutant CLL and HCC tumors shown a clear enrichment of G at the +6 position that is not observed in canonical or inefficiently spliced 5′ splice sites in mutant tumors [18]. Similarly, the cryptic 5′ splice sites observed in MBL harboring the 3A > G mutation are enriched in C at the +6 position [19]. As a consequence of the mismatch with the canonical sequence, intron retention was observed in MBL tumors harboring the U1 snRNA A > G mutation [19]. Indeed, 5′ splice sites from the retained introns are enriched in uracil at +6 position, the preferred 5′ splice site consensus motif that is well recognized by wild-type U1 snRNA. This observation suggests that, at least in MBL, mutant U1 snRNA could inhibit the canonical recognition by wild-type U1 snRNA on some specific 5′ splice sites [19]. However, how mutant U1 snRNA inhibits wild-type U1 snRNA recognition and why only a subset of 5′ splice sites are subjected to this inhibition remain elusive.

### 2.2. Oncogenic Roles of Mutant U1 snRNA 3A > C/G

One of the important questions that remains unanswered is whether the mutations described above are required/sufficient to drive tumor onset or sustain tumor progression. Important clues about the oncogenic potential of U1 snRNA mutations came from transcriptome analysis of U1 snRNA mutant tumors that revealed oncogenic programs potentially regulated by U1 snRNA mutations [18]. CLL tumors harboring the 3A > C mutation showed the up-regulation of genes belonging to RNA processing (transcription and splicing), protein ubiquitination, and telomere maintenance, whereas genes belonging to apoptosis, B-cell receptor signaling, and cytoplasmic ribosome processes were down-regulated. Moreover, splicing pattern analysis of the tumors identified aberrant splicing events in genes known to be involved in cancer, such as CD44, POLD1, MSI2, and ABCD3. In the case of SHH medulloblastomas, the U1 snRNA A > G mutation was associated with splicing alterations in components of the sonic hedgehog signaling pathway, such as PTCHI, GLI2, and CCND2 [19]. This observation is relevant as sonic hedgehog signaling activation is a known driver of SHH medulloblastoma [23], thus directly associating U1 snRNA mutations with SHH tumorigenesis. Detailed functional in vivo analysis of the contribution of mutant U1 snRNA and the role of the generated cancer isoforms is required to uncover the oncogenic role of the mutant U1 snRNA and its effectors, which will be instrumental to exploit novel therapeutic strategies.

## 3. Spliceosomes Containing Mutant SF3B1 Promote Tumorigenesis through the Use of Cryptic 3′ Splice Sites

One of the most exciting discoveries of the recent years has been the recurring somatic mutations in SF3B1, in particular in hematopoietic and lymphoid malignancies [24]. SF3B1 is an essential component of the U2 snRNP, which is critical to BPS recognition and essential for 3′ splice site selection [1,2]. SF3B1 mutations are highly prevalent in myelodysplasia syndromes (MDS, ~30%), reaching mutational rates up to 83% in the refractory anemia with ring sideroblasts MDS subtype (RARS) [25]. SF3B1 is also the most frequently mutated gene in CLL (~14%) [26,27], and one of the few genes found to be frequently mutated in uveal melanoma (UVM) (~18%) [28]. Interestingly, SF3B1 mutations correlate with good prognosis in MDS and UVM patients [28,29], while a reduced overall survival was observed in CLL and luminal B and progesterone receptor-negative breast cancer patients [30,31,32].

Alterations of the SF3B1 gene, such as non-silent mutations, could alter both splice site recognition and splicing decision. Indeed, SF3B1 harbors unique characteristics that support its function. On the one hand, SF3B1 acts on splice site recognition, crosslinking with RNA at both flanking sites of the BPS [7]. On the other hand, SF3B1 can directly interact, via its N-terminal region, with the BPS binding protein and U2 snRNP subunit p14 [33] and with the PPT binding protein U2AF2 [7].

Remarkably, mutations in SF3B1 are heterozygous and the vast majority are single nucleotide missense mutations located at the C-terminal of the protein inside the HEAT-repeat domain (HD) (Figure 3A). The most frequent SF3B1 mutations are K700E, K666N, and R625H. The K700E mutation is the prevalent mutation in MDS and CLL [25], whereas mutations in R625 are highly frequent in UVM [28]. Hereafter, these hotspot mutations will be referred to as SF3B1^MUT^.

### 3.1. Molecular Basis of Cryptic 3′ Splice Site Usage by SF3B1^MUT^ Spliceosomes

Transcriptome analysis of SF3B1^WT^ and SR3B1^MUT^ tumors from CLL, UVM, and breast cancer patients allowed the identification of splicing alterations associated with SF3B1 hotspot mutations and revealed that SF3B1^MUT^ stimulates the usage of cryptic 3′ splice sites by recognizing a cryptic BPS [34,35,36]. Indeed, the most frequently altered splicing events in SF3B1^MUT^ tumors are cryptic 3′ splice site usage (up to 60–76%) [34,35,36], with 3′ cryptic splicing sites located both close to (proximal, ~55%) or far away from (distal, ~45%) the canonical 3′ splice site [34].

**Figure 3 cancers-14-00281-f003:**
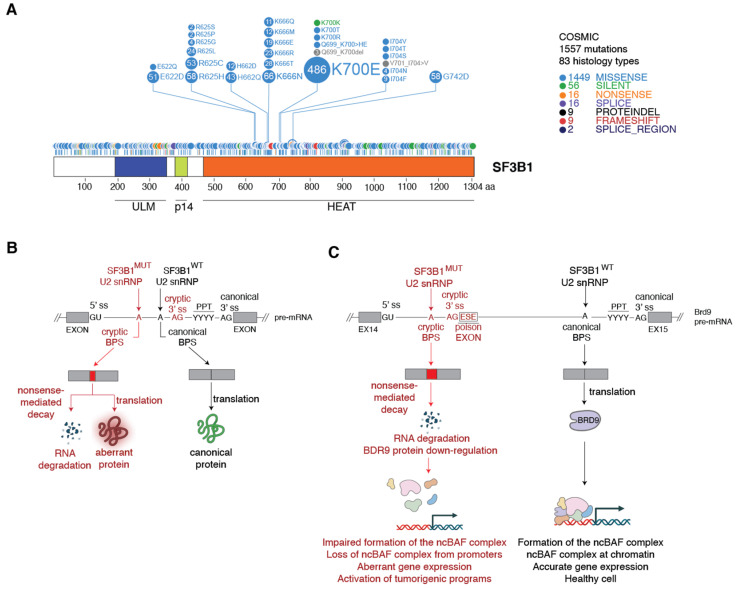
Mutant SF3B1 binds cryptic BPS and promotes the use of cryptic 3′ splice sites. (**A**) Schematic representation of the SF3B1 protein domains and its mutation profile. SF3B1 domains are indicated: U2AF-homology ligand motif (ULM), p14 binding domain (p14), and HEAT-repeat domain (HEAT). The mutation profile was obtained from the COSMIC database (https://cancer.sanger.ac.uk/cosmic, accessed on 25 October 2021) and visualized using ProteinPaint (https://pecan.stjude.cloud/proteinpaint, accessed on 25 October 2021). The number of samples harboring each mutation is indicated inside each circle and their relative abundance is represented by the disc size. (**B**) Sequence requirements for cryptic BPS usage by mutant SF3B1. A schematic exon-intron-exon region is shown together with the characteristic sequence features present in the splicing events altered by SF3B1^MUT^: the 5′ splice site (5′ ss), the canonical and cryptic 3′ splice sites (3′ ss), the canonical and cryptic BPS, and the PPT. Red indicates the SF3B1 mutant, the adenine at the cryptic BPS, and the AG dinucleotide at the 3′ cryptic splice site. (**C**) Mutant SF3B1 promotes the inclusion of a poison exon in the BRD9 transcript. Schematic representation of the BRD9 pre-mRNA containing the exon 14-intron-exon 15 region. The required splicing sequences are indicated as in (**B**). ESE, exon splicing enhancer. Poison exon inclusion targets the BRD9 transcript to NMD, promoting BRD9 protein down-regulation and impaired formation of the ncBAF complex.

Analysis of specific splicing events in model cell lines showed that neither SF3B1 knock-down (KD) nor SF3B1 over-expression recapitulate the splicing alterations observed in SF3B1^MUT^ cells, indicating that hotspot mutations in SF3B1 are likely change-of-function mutations [36]. The main consequence of the cryptic 3′ splice site usage by SF3B1^MUT^ is the loss of the open reading frame, with ~50% of the affected transcripts being targeted for degradation by non-sense mediated decay (NMD), leading to reduced expression of the codified protein [35] (Figure 3B).

Proximal cryptic 3′ splice sites recognized by SF3B1^MUT^-U2 snRNPs are characterized by a sequence signature that includes a cryptic AG dinucleotide (cryptic 3′ spice site) localized upstream the canonical 3′ splice site and downstream the canonical BPS [35,36] (Figure 3B). In addition, a hidden BPS, rich in adenines, is present upstream the cryptic AG 3′ splice site. SF3B1^MUT^-U2 snRNP recognizes an adenine present in this cryptic BPS, thus stimulating the usage of the cryptic 3′ splice site [35,36].

Interestingly, SF3B1^MUT^ likely stimulates the usage of distal and proximal 3′ splice sites by the very same mechanism [34]. Indeed, inspection of the distal cryptic 3′ sites exploited by SF3B1^MUT^ led to the identification of an upstream adenosine-rich sequence, very similar to the cryptic BPS sequence used by SF3B1^MUT^ at proximal 3′ splice sites. As not all introns contain such cryptic BPS required for SF3B1^MUT^ recognition, only a subset of transcripts is actually aberrantly spliced in SF3B1^MUT^ tumors.

Why the SF3B1^MUT^-U2 snRNP binds preferentially to a cryptic BPS instead of the canonical one remains unclear. It has been proposed that SF3B1^MUT^ may possess an enhanced affinity towards the specific nucleotides at flanking regions of the cryptic BPS [35]. Another hypothesis suggests that SF3B1^MUT^ could induce a conformational change in the U2 snRNP, which in turn promotes binding to the cryptic BPS [36]. Unfortunately, at present, no data validated these models. Using purified SF3b complexes and crosslinking experiments, SF3B1^WT^ and SF3B1^MUT^ were shown to bind equally efficiently to a synthetic BPS-PPT RNA [37]. Moreover, SF3B1^MUT^ did not seem to affect the structure and molecular interaction of the SF3b complex [37], including the interaction with U2AF2 in vivo [36] and in vitro [37]. Thus, SF3B1^MUT^ shows no major effects on the topological architecture of the splicing complex which, however, does not exclude minor but critical conformational changes in the context of a full U2 snRNP complex where the stability of the U2 snRNP:BPS interaction could be affected by SF3B1 mutations.

A third possibility is that SF3B1 may affect binding with interactors outside the U2 snRNP, such as the intron itself or additional splicing factors. Indeed, the hotspot residue K700 has been found to be exposed to the solvent in the SF3b complex structure [37]. By purifying and analyzing the composition of U2 snRNP complex wild-type and mutant from K562 myelogenous leukemia cells, Zhang and colleagues found that the SUGP1 protein is lost in SF3B1^MUT^-containing complexes [38]. Strikingly, SUGP1 KD in SF3B1^WT^ cell lines recapitulates the usage of cryptic BPS and cryptic 3′ splice sites observed in SF3B1^MUT^ cell lines [38]. Furthermore, the over-expression of SUGP1 in SF3B1^MUT^ cells rescues the interaction between SUGP1 and SF3B1^MUT^-containing complexes and partially rescue 3′ splice site usage alterations [38]. Both observations suggest that the loss of SUGP1 from SF3B1^MUT^ spliceosomes could dictate the usage of the cryptic 3′ splicing site observed in SF3B1^MUT^ tumors.

How SUGP1 achieves such a critical role remains an open question. Interestingly, point mutations in the G-patch domain at the C-terminal of SUGP1 is sufficient to induce the usage of cryptic 3′ splice sites in cell lines [38]. Based on the role of G-patch proteins as activators of DEAH-box RNA helicases [39], it has been proposed that SUGP1 could act by modulating RNA helicase activity during splicing [38]. By doing so, SUGP1 could mediate the remodeling and exchange of splicing factors at the conventional BPS, such as the displacement of SF1 by the U2 snRNP. Thus, in the context of the SF3B1^MUT^, SUGP1 would not be recruited at the spliceosome, forcing U2 snRNP deposition at the cryptic BPS [38]. Again, this intriguing model lacks experimental validation, as the hypothetical RNA helicase modulated by SUGP1 remains unsettled.

### 3.2. Mechanism of SF3B1^MUT^-Driven Tumorigenesis

SF3B1 mutations have multiple consequences in tumorigenesis. It has been suggested that SF3B1 mutations arise later in leukemia development and contribute to disease progression [27]. Accordingly, mutations in SF3B1 correlate with a rapid disease progression in CLL [40]. Mice expressing heterozygous and hematopoietic-restricted SF3B1 K700E mutation do not develop leukemia but exhibit macrocytic anemia due to a block in terminal erythroid maturation, erythroid dysplasia, and long-term hematopoietic stem cell expansion, all of which are characteristic features of MDS [15]. Transcriptome analysis of CLL samples from SF3B1^MUT^ tumors, identified expression alterations in genes belonging to biological pathways associated with poor prognosis [41]. An example is represented by the Notch pathway, whose aberrant signaling has been linked to CLL with reduced overall survival [42]. Myeloid and leukemic cell lines expressing SFR3B1^MUT^ exhibit a higher Notch pathway activity [26], possibly due to the expression of a DVL2 exon 11-aberrant isoform unable to repress Notch signaling [41]. Correlative evidence was also found for a few altered splice variants induced by SF3B1^MUT^ (CHD1L, GAK, RAD9A, JMY) involved in the DNA damage response [41,43].

Recently, Inoue and colleagues identified one key oncogenic effector of the SF3B1^MUT^ spliceosome, the non-canonical chromatin remodeling BAF complex (ncBAF) subunit BRD9 [17]. By using a distal cryptic 3′ splice site, a cryptic BPS, and an exon splicing enhancer (ESE), SF3B1^MUT^ induces the inclusion of a new exon upstream the BRD9 exon 15 (Figure 3C). This poison exon (14a) interrupts the open reading frame and targets the aberrant BRD9 transcript to NMD, thereby down-regulating BRD9 expression. Remarkably, the inclusion of exon14a was observed in all cohorts of CLL, UVM, and MDS patients harboring SF3B1 hotspot mutations [17]. At the functional level, SF3B1^MUT^, by inducing BRD9 loss, disrupts the ncBAF complex and, as a consequence, its loss from chromatin affects the expression of genes involved in apoptosis, cell growth, adhesion, and migration [17]. Validating experiments showed that BRD9 KD in non-tumorigenic Melan-a melanocytes results in tumor growth and the induction of melanomagenesis in vivo while BRD9 depletion in melanoma cells increases the number of pulmonary metastases [17]. Finally, the inhibition of BRD9 poison exon inclusion by using specific antisense oligonucleotides (ASOs) rescues BRD9 expression in SF3B1^MUT^ melanoma cells in vitro and inhibits tumor growth by inducing tumor necrosis in vivo [17]. Thus, the loss of BRD9 represents the first SF3B1^MUT^-induced pro-oncogenic event driving tumorigenesis. While more in-depth functional studies are needed to uncover further critical oncogenic effectors of SF3B1^MUT^, these studies have validated ASOs as a valuable therapeutic option to treat SF3B1^MUT^ cancers.

## 4. Cancer-Associated U2AF1 Mutations Influence U2AF1 3′ Splice Site Recognition and Splicing Outcome

A third cancer-associated mutated splicing component is U2AF1, which is recurrently mutated in myelodysplasia syndromes (MDS, ~9%) and hematological malignancies such as chronic myelomonocytic leukemia (CMML, ~8%) and acute myeloid leukemia (AML, ~4%) [25,44,45]. U2AF1 mutations are more frequent in high-risk MDS patients and are associated with reduced survival in CMML patients [45]. Similar bad prognosis and higher frequency of disease persistence after chemotherapy were observed in intermediate-risk AML patients with U2AF1 mutations [46]. Finally, U2AF1 mutations are present in lung adenocarcinoma (LUAD, ~5%), where they are associated with reduced patient survival [35,47].

From the molecular point-of-view, U2AF1 heterodimerizes with U2AF2, forming the U2 snRNP auxiliary factor U2AF, which is essential for 3′ splice site recognition [5,6]. U2AF1 specifically recognizes the AG dinucleotide at the 3′ splice site [48,49,50], whereas U2AF2 binds the PPT and interacts with the U2 snRNP [7,50] (Figure 1A).

In this case also, U2AF1 mutations are heterozygous and the retained wild-type allele is expressed, suggesting that mutant cancer cells may require the residual normal protein to be viable. Remarkably, the mutations are mostly located at two hotspot positions, Ser34 (S34F/Y) and Glu157 (Q157P/R) (Figure 4A). S34F/Y is the most recurrent U2AF1 mutation in MDS and AML, and is the unique mutation found in LUAD [25,44,51,52]. The two cancer-associated residues map in the two zinc finger domains (ZnF) present in U2AF1 (Figure 4A). Recent structural studies performed with fission yeast U2AF1 in complex with a 3′ splice site RNA sequence suggested that both ZnFs bind and recognize the AG dinucleotide [53]. This biochemical study has illuminated the molecular mechanism of the recognition of 3′ splice site by U2AF1 suggesting a dysfunctional role of U2AF1^MUT^ in RNA recognition and, as a consequence, in the splicing outcome.

### 4.1. Molecular Basis of Altered 3′ Splice Site Recognition by Mutant U2AF1

Deep analysis of the transcriptome from patient-derived primary samples (MDS, AML, LUAD) and human and mouse cell lines expressing U2AF1^WT^ or U2AF1^MUT^ have allowed the identification of rare U2AF1^MUT^-dependent splicing alterations [16,51,52,54,55,56]. In agreement with its moderate effect on splicing, iCLIP data from HCC78 LUAD cells ectopically expressing U2AF1^WT^ or U2AF1^S34F^ showed only ~20% difference in occupancy between U2AF1^WT^ and U2AF1^S34F^ [56]. The most common splicing events altered in U2AF1^MUT^ cancers are exon cassettes (~40–60%) followed by alternative 3′ splice sites usage (~20%).

In contrast to SF3B1 mutations, U2AF1^MUT^ does not feed the NMD pathway with new substrates [54], but alters both exon usage and polyadenylation site selection and translation efficiency by de-regulating U2AF1 binding to polyadenylation sites and 5′ UTR regions, respectively [57,58]. Interestingly, the S34F/Y and Q157P/R mutations present different patterns of exon inclusion [54], indicating a differential impact on cancer cell physiology. U2AF1^MUT^-regulated exons are characterized by specific sequence features. Exons included by the U2AF1^S34F/Y^ mutant are enriched in C/A nucleotide at position -3, whereas frequently skipped exons often have a U in the same position [16,51,52,54,55,56] (Figure 4B). Importantly, adjacent exons do not show such enrichment and have an equal probability of C and U enrichment at the -3 position [51]. In line with the sequence analysis, iCLIP data from HCC78 LUAD cells, ectopically expressing U2AF1^WT^ or U2AF1^S34F^, demonstrated that U2AF1^S34F^ preferentially binds (C/A)AG over UAG [56]. This sequence signature was also observed inside exon sequences subjected to 3′ alternative splice site usage by mutant U2AF1^S34F/Y^ [52,55] (Figure 4B).

In the case of the U2AF1^Q157P/R^ mutant, preferentially included exons were enriched in G over A at the +1 nucleotide [54] (Figure 4B). The same splice site sequence selectivity was also observed in K562 erythroleukemic cell lines ectopically expressing U2AF1^WT^, U2AF1^S34F/Y^ or U2AF1^Q157P/R^ [54]. Importantly, splicing events altered by U2AF1 KD in this cellular model did not show any particular sequence enrichment, and did not mimic U2AF1^MUT^. Thus, splicing alterations observed in U2AF1^MUT^ tumors are the consequence of change-of-function mutations [54], with the U2AF1^S37F/Y^ mutant preferentially splicing (C/A)AG over UAG splice sites (Figure 4B). It is worth mentioning that the S37F/Y and Q157P/R mutations have not been investigated at the structural and biochemical level and future molecular studies on U2AF1^MUT^:RNA interaction are predicted to shed light on the mechanism underlying U2AF1 activity.

### 4.2. Consequences of U2AF1 Mutations on Tumorigenesis

Transgenic mice expressing the U2AF1^S37F^ mutation display various abnormalities associated with MDS, including peripheral blood leukopenia, reduction in B cells and monocytes in the bone marrow, and bone marrow progenitor cell expansion [16,20]. In addition, transcriptome analysis of common myeloid progenitors (CMPs) from transgenic mice identified U2AF1^S34F^-dependent gene expression alteration in genes belonging to immune response and leukocytic activation processes [16]. Nevertheless, U2AF1^S34F^ transgenic mice do not develop MDS or AML [16]. Although U2AF1^MUT^ does not generate widespread splicing alterations, altered splicing events driven by U2AF1^MUT^ belong to mutated genes in MDS and AML (ASXL1, GNAS, PICALM) and involved in cancer hallmarks, including stem cell biology (MED24), transcription regulation (H2AFY), replication stress response pathway (ATR), and innate immune pathway (IRAK4) [16,20,54,59].

The defects in B cell development observed in U2AF1^S34F^ transgenic mice [20] have been associated with U2AF1^S34F^-mediated splicing alterations of the transcript of histone variant H2AFY (macroH2A1), which is associated with both transcription repression and transcription activation [60]. Two different isoforms are generated by the mutually exclusive inclusion of exon 6a and exon 6b, leading to H2AFY1.2 and H2AFY1.1, respectively (Figure 4C). Wild-type human MDS bone marrow expresses both isoforms, while U2AF1^S34F/Y^ MDS samples express only the H2AFY1.2 isoform, indicating that U2AF1S^34F/Y^ promotes exon 6b skipping and H2AFY1.1 down-regulation [16] (Figure 4C). Sequence feature analyses support this idea: exon 6a presents a CAG trinucleotide at the 3′ splice site, whereas exon 6b presents a UAG trinucleotide, characteristic sequences for U2AF1^S34F/Y^ exon inclusion and exon skipping, respectively (Figure 4C).

Interestingly, H2AFY knock-out (KO) mice present the same B cell development defects, but not the additional MDS features observed in U2AF1^S34F^ transgenic mice [20]. Moreover, the expression of H2AFY1.1, but not H2AFY1.2, rescues the B cell development defects observed in both H2AFY KO and U2AF1^S34F^ transgenic mice [20]. These results mechanistically link the U2AFY splicing alteration to the defects in B cell development observed in U2AF^S34F^ transgenic mice. Molecularly, the down-regulation of the U2AFY1.1 isoform leads to its loss at the promotor of the transcription factor EBF1, which is critical for B cell development, and is, therefore, the key responsible for the phenotype observed [20].

Another oncogenic effector of U2AF1^MUT^ is the interleukin-1 receptor-associated kinase 4 (IRAK4) (Figure 4D). IRAK4 is a Serine/Threonine kinase that acts downstream the Toll-like receptor superfamily, activating NF-κB and MAPK signaling in inflammation [61]. In AML, U2AF1^MUT^ mediates the inclusion of IRAK4 exon 4 and, therefore, the expression of the IRAK4 long isoform (IRAK4-L), which is frequently expressed in AML patients with poor outcome [59]. The recognition of the 3′ splice site at exon 4 by U2AF1^MUT^ is favored by the presence of an adenine at the -3 position and a guanine at the +1 position, which are the preferred binding splice site sequences for U2AF1^S34F/Y^ and U2AF1^Q157P/R^, respectively (Figure 4D). Molecularly, exon 4 codifies for the N-terminal death-domain (DD) that is essential for protein–protein interactions and the formation of the myddosome signaling complex. In AML, the U2AF1^MUT^-mediated expression of IRAK4-L stabilizes IRAK4 interaction with myddosome components, resulting in maximal activation of NF-κB signaling, which is essential for leukemic cell fitness [59]. Importantly, in vivo xenografts from MDS patients harboring U2AF1 mutations were strongly affected when treated with the IKAR4 inhibitor CA-4948, showing a 50% decrease in MDS cell engraftment. These observations uncover a previously unknown vulnerability to IRAK4 inhibitors of AML tumors harboring U2AF1 mutations, which might be relevant for therapeutical approaches such as targeted ASOs.

## 5. Conclusions and Outlook

Genome-wide transcriptome analysis identified 119 splicing factor genes with non-silent mutations across different tumors, including hotspot and loss-of-function mutations [24]. Tumors appear to modify the splicing outcome by selecting alterations in key factors involved in early stages of the splicing process. Indeed, hotspot mutations in the U2 spliceosome components SF3B1, U2AF1 [24], and U1 snRNA [18,19] were found to be exceptionally well represented. On the other hand, to restrict the otherwise broad and deleterious effects generated by defects on splice site selection, cancer cells appear to select specific mutations to achieve a “balanced alteration”. This is ensued in two ways: first, U1 snRNA, SF3B1, and U2AF1 mutations occur in specific hotspot bases/residues that restrict their binding and activity to a specific group of transcripts; second, the heterozygous nature of the mutations guaranties the presence of the wild-type spliceosomes, which can carry out splicing on canonical splice sites. Cells cannot tolerate a dramatic and widespread alteration in splicing, as demonstrated by the fact that spliceosome mutations are mutually exclusive [18,25]. In addition, cancer cells with SF3B1 or U2AF1 mutations require the presence of the wild-type spliceosome to survive [16,62,63]. Thus, cancer cells exploit spliceosome mutations to drive cancer-specific and tolerable splicing changes in order to support transformation and tumor development.

Importantly, SF3B1^MUT^ or U2AF1^MUT^ cancer cells are highly susceptible to pharmacology-driven splicing perturbations [15,16], a phenomenon that has been therapeutically exploited. Unfortunately, the failure of the phase I clinical trial of the SF3B1 inhibitor E7107 in solid tumors [64] has underlined the need for specific therapies that are non-toxic for normal cells. A more selective strategy is represented by ASOs, which, by targeting specific splicing events, allow the restoration of non-oncogenic isoforms [65]. The use of nusinersen in the treatment of patients with spinal muscular atrophy is paradigmatic of this therapeutic approach [66]. Preclinical work on the SF3B1 target BRD9 suggests that ASOs might also have a therapeutic value in oncology [17]. A similar strategy could be exploited to restore H2AFY1.1 expression and inhibit IRAK4-L expression in CLL U2AF1^S34F^ cells. It is predicted that uncovering additional oncogenic isoform effectors of the mutant spliceosome will not only improve our understanding of how the mutant spliceosome drives tumor development but will also allow the design of more selective therapeutic strategies.

## Figures and Tables

**Figure 1 cancers-14-00281-f001:**
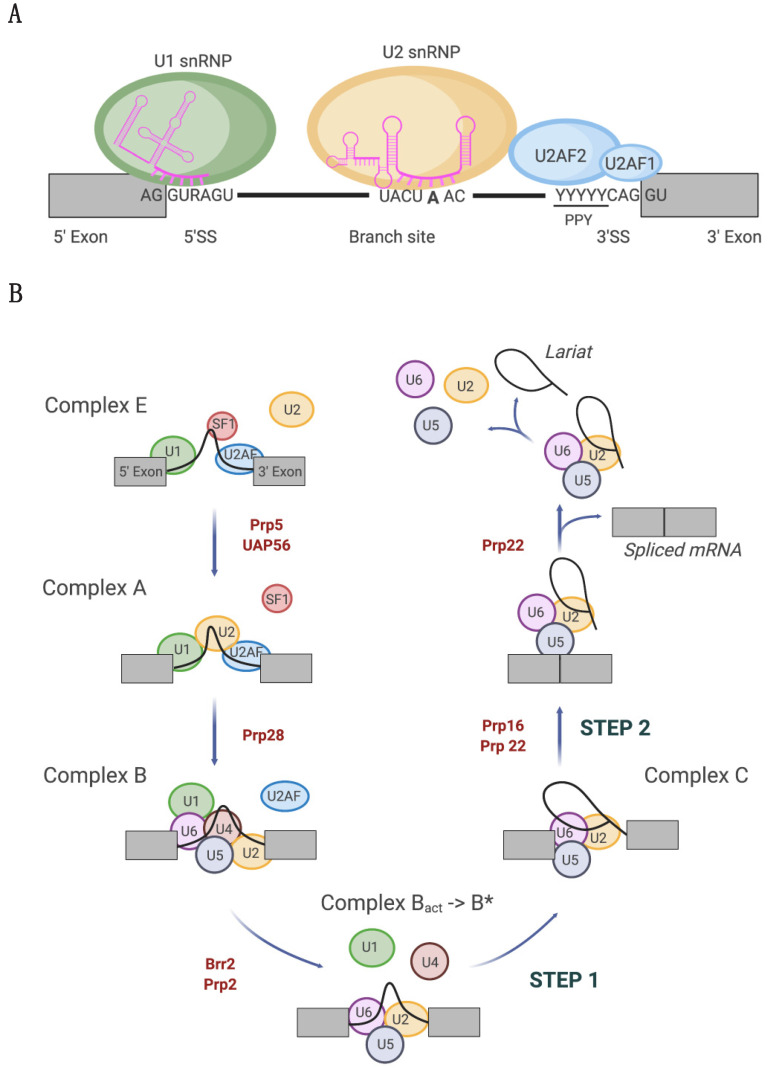
The splicing cycle. (**A**) Splicing complex assembly is initiated by the recognition of consensus sequence elements located at the exon (grey boxes)/intron (black line) boundaries. U1 snRNP and U2AF recognize the 5′ (5′ ss) and 3′ (3′ ss) splice sites, respectively, inducing U2 snRNP recruitment at the branch point. (**B**) Pre-mRNA splicing stepwise reaction performed by the spliceosome is depicted. U1, U2, U4, U5, and U6 subunits are composed of snRNAs and Sm/LSm proteins. Details of the process are given in the main text. Rearrangements and remodeling are assisted by ATPases/helicases (indicated in red) to allow the progression through complexes E, A, B, B*, and C.

**Figure 2 cancers-14-00281-f002:**
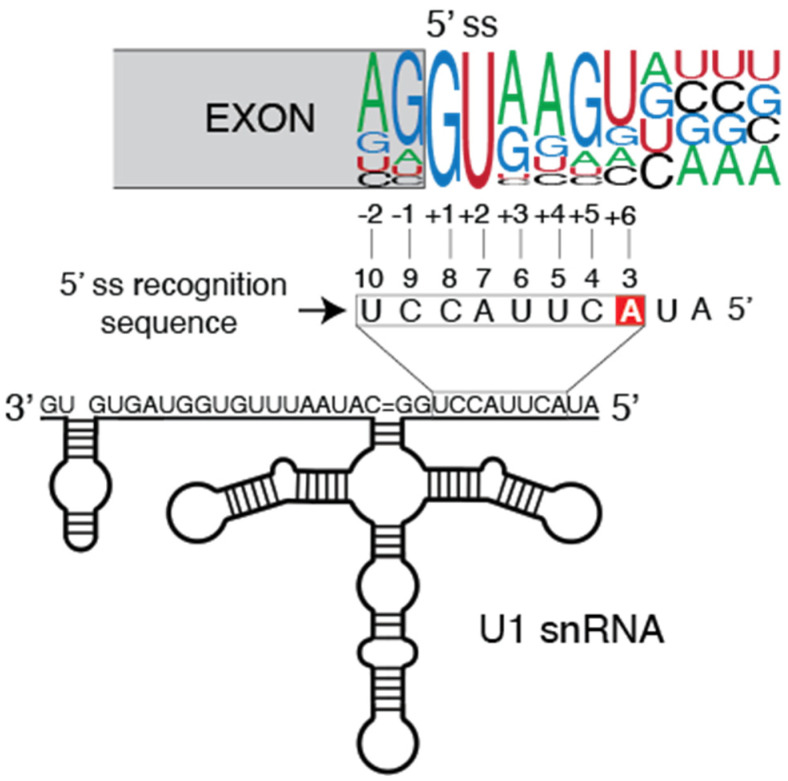
Canonical U1 snRNA and 5′ splice site base-pairing. Schematic representation of the base-pair interaction between the U1 5′ splice site recognition sequence (box) and the 5′ splice site consensus motif. The numbers above the box indicate the base position with respect to the 5′ end. The adenine at the third base of the U1 snRNA, which is recurrently mutated in cancer, is highlighted in a red box. The 5′ splice site consensus motif is shown and the numbers below the sequence indicate the base position at the exon (−) and the intron (+) with respect to the exon/intron junction.

**Figure 4 cancers-14-00281-f004:**
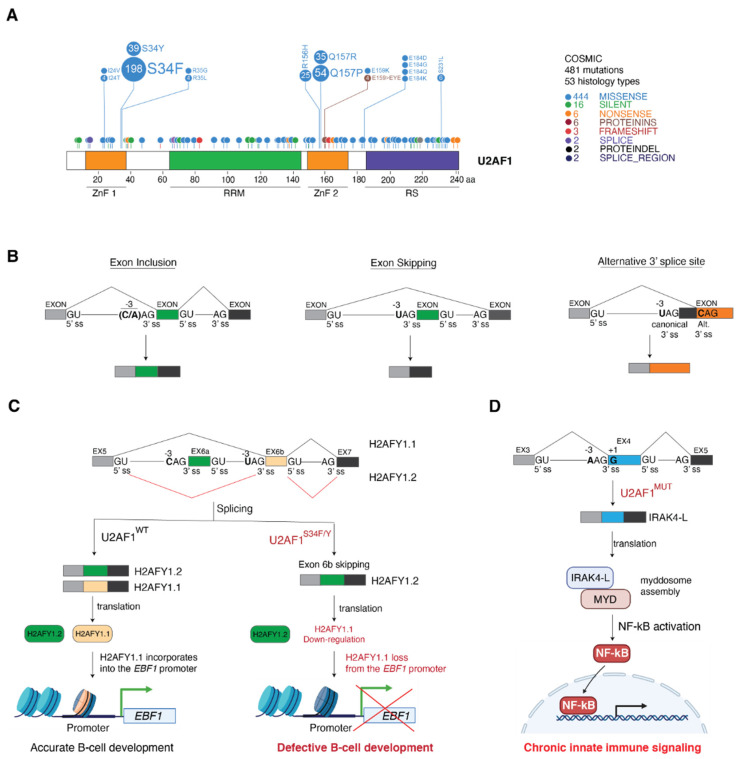
U2AF1 mutations influence U2AF1 3′ splice site recognition and splicing outcome. (**A**) Schematic representation of the U2AF1 protein with its mutation profile. U2AF1 domains are indicated: zinc finger domains 1 (ZnF 1) and 2 (ZnF 2), the RNA recognition motif (RRM) and the arginine/serine rich domain (RS). The mutation profile was obtained from the COSMIC database (https://cancer.sanger.ac.uk/cosmic, accessed on 25 October 2021) and visualized using ProteinPaint (https://pecan.stjude.cloud/proteinpaint, accessed on 25 October 2021). The number of samples harboring each mutation is indicated inside each circle and their relative abundance is represented by the disc size. (**B**) Features of splicing events affected by the S34F/Y U2AF1 mutant. Schematic exon-intron-exon regions are shown together with the characteristic sequence features present in the splicing events altered by U2AF1^S34F/Y^: exon inclusion (left), exon skipping (center), and alternative 3′ splice site usage. The positions of the 5′ splice site (5′ ss), the canonical 3′ splice sites (3′ ss), and the alternative 3′ splice site (Alt 3′ ss) are indicated. The -3 nucleotide, which impacts on 3′ splice site selection by mutant U2AF1^S34F/Y^, is highlighted in bold. (**C**) U2AF1^S34F/Y^ down-regulates the H2AFY1.1 transcript, leading to B-cell developmental defects. Schematic representation of the H2AFY pre-mRNA from exon 5 to exon 7, containing the mutually exclusive exons 6a and 6b. Splicing sequences are indicated as in (**B**). Both H2AFY isoforms are expressed in U2AF1^WT^ cells. In U2AF1^S34F/Y^ mutant cells, exon 6b is excluded; thus, H2AFY1.1 is not expressed. H2AFY1.1 loss down-regulates *Ebf1* expression and leads to B-cell developmental defects. (**D**) Mutant U2AF1 up-regulates IRAK4 long isoform expression and NF-kB activation. Schematic representation of the IRAK4 pre-mRNA from exon 3 to exon 5, containing the alternative splice exon 4. Splicing sequences are indicated as in (**B**). The +1 G base is indicated in bold. IRAK4-L promotes the myddosome complex (MYD) assembly and activates NF-kB signaling.

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
