# Peer review of "Recurrent Spliceosome Mutations in Cancer: Mechanisms and Consequences of Aberrant Splice Site Selection"

_cancers, 2022, doi:10.3390/cancers14020281_

Round 1

Reviewer 1 Report

This work presents an intensive review of U1 snRNA, SF3B1 and U2AF1 59
somatic mutations and the mechanism by which they influence splicing. However, the paragraphs is hard to follow.  What is the main message or purpose of this review? What is the current gap in evidence need to be bridged or future direction of this filed? Limitation?  Without a clear aim, the massive literature review is a bit out of focus. 

Author Response

The discovery of oncogenic mutations in splicing factors and the growing evidence of aberrant splicing in various cancers, highlight RNA splicing as critical hallmark in oncogenesis. This review intends to discuss the most recent findings, namely the discovery of  recurrent hot spot mutations in the spliceosome components U1 snRNA, SF3B1 and U2AF1 that have been identified across different tumor types. We extensively reviewed the manuscript in order to better convey this message.

Reviewer 2 Report

The review article " Recurrent spliceosome Mutations in Cancer: Mechanisms and Consequences of Aberrant Splice Site Selection" is nicely described and can be valuable for the reader.

However I have some comments described below:

  1. Too many abbreviations, it will be nice if abbreviations are described in the beginning when it is used first.
  2. Explain Figure legend in details like in Fig 1: Complex B act to * , it will be easy to follow if act and *  are described.
  3. Check for references, mention wherever needed. 

Author Response

We thank the Reviewer for his/her nice words. We apologise for not having explained the Figures in details. We have now reviewed their legends and added the list of abbreviations, as suggested. In addition, we further checked the list of references.

Reviewer 3 Report

Defects in alternative splicing are found in human tumors due to mutations in the splicing regulatory elements of cancer-specific genes or due to changes in regulatory splicing mechanisms.

RNA splicing regulators constitute a new class of oncoproteins and tumor suppressor genes and contribute to disease progression by modulating RNA isoforms involved in cancer signaling pathways. The transcriptional and post-transcriptional regulation of splicing factors modulates their levels and activities in tumor cells. Mutations in pre-mRNA regulatory sequences, splicing regulators, and chromatin modifiers, as well as differential expression of splicing factors, are essential contributors to cancer pathogenesis. It has become clear that these aberrations contribute to many facets of cancer, including oncogenic transformation, cancer progression, response to anticancer drug treatment, and resistance to therapy.

The mutated RNA splicing mechanism leads to many diseases and is a promising therapeutic target for small molecule treatment. In the case of mutations in individual genes that cause incorrect splicing, engineered splicing factors can be introduced to correct the splicing of these aberrant transcripts and thus reduce the effects of disease phenotype expression.

Cancer cells with global splicing alterations are dependent on transcriptional products derived from the wild spliceosome for their survival, which potentially creates a therapeutic vulnerability in cancers with a mutant spliceosome. These small molecules that target splicing factors have been developed to induce apoptosis in cancer cells preferentially.

Current efforts are being made to block alternative splicing, including global splicing inhibition using small molecules that block spliceosome or splicing factor-modifying enzymes, as well as splice-switching RNA-based therapeutics to modulate splice-specific splicing isoforms.

The discovery of oncogenic mutations in splicing factors, and growing evidence of disturbed splicing in various cancers, highlight RNA processing defects as a critical driver of oncogenesis. Aberrant splicing is an important source of new cancer biomarkers, and the spliceosome functional mechanism represents an attractive target for a rapidly expanding new class of therapeutic agents. These findings have resulted in a growing interest in targeting RNA splicing as a therapeutic approach to cancer treatment.

Some aspects are still not completely clarified. It is unclear how and why the ubiquitously expressed variants in spliceosome proteins required for pre-mRNA splicing in all human cells result in disease phenotypes restricted to only a single tissue. Moreover, it is unclear why the different interacting protein variants that compose the same snRNP of the central spliceosome result in a completely different disease.

Author Response

We thank the Reviewer for his/her nice words that perfectly summarised what we intended to highlight with this review.